# Suicide after leaving the UK Armed Forces 1996–2018: A cohort study

Cathryn Rodway[1]*, Saied Ibrahim[1], Jodie Westhead[1], Lana Bojanić[1], Pauline Turnbull[1], Louis Appleby[1], Andy Bacon[2,3], Harriet Dale[4], Kate Harrison[4], Nav Kapur[1,5,6]

1 National Confidential Inquiry into Suicide and Safety in Mental Health (NCISH), Centre for Mental Health and Safety, School of Health Sciences, University of Manchester, Manchester, United Kingdom, 2 Armed Forces Team, NHS England, London, United Kingdom, 3 Westminster Centre for Research in Veterans, University of Chester, Chester, United Kingdom, 4 Ministry of Defence, Defence Statistics Health, Bristol, United Kingdom, 5 NIHR Greater Manchester Patient Safety Research Centre, University of Manchester, Manchester, United Kingdom, 6 Mersey Care NHS Foundation Trust, Liverpool, United Kingdom

* cathryn.a.rodway@manchester.ac.uk

## Abstract

### Background

There are comparatively few international studies investigating suicide in military veterans and no recent UK–wide studies. This is important because the wider context of being a UK Armed Forces (UKAF) veteran has changed in recent years following a period of intensive operations. We aimed to investigate the rate, timing, and risk factors for suicide in personnel who left the UKAF over a 23–year period.

### Methods and findings

We carried out a retrospective cohort study of suicide in personnel who left the regular UKAF between 1996 and 2018 linking national databases of discharged personnel and suicide deaths, using survival analysis to examine the risk of suicide in veterans compared to the general population and conditional logistic regression to investigate factors most strongly associated with suicide after discharge. The 458,058 individuals who left the UKAF accumulated over 5,852,100 person years at risk, with a median length of follow–up of 13 years, were mostly male (91%), and had a median age of 26 years at discharge. 1,086 (0.2%) died by suicide. The overall rate of suicide in veterans was slightly lower than the general population (standardised mortality ratio, SMR [95% confidence interval, CI] 94 [88 to 99]). However, suicide risk was 2 to 3 times higher in male and female veterans aged under 25 years than in the same age groups in the general population (age–specific mortality ratios ranging from 160 to 409). Male veterans aged 35 years and older were at reduced risk of suicide (age–specific mortality ratios 47 to 80). Male sex, Army service, discharge between the ages of 16 and 34 years, being untrained on discharge, and length of service under 10 years were associated with higher suicide risk. Factors associated with reduced risk included being married, a higher rank, and deployment on combat operations. The rate of contact with specialist NHS mental health services (273/1,086, 25%) was lowest in the

**Data Availability Statement:** Data cannot be shared publicly because of information governance restrictions in place to protect confidentiality. Access to NCISH data can be requested via application to the Healthcare Quality Improvement

Partnership (www.hqip.org.uk/national-programmes/accessing-ncapop-data/).

**Funding:** CR, SI, JW, LB, and NK received funding for this study jointly from the UK Ministry of Defence (MOD; https://www.gov.uk/government/organisations/ministry-of-defence) and NHS England (https://www.england.nhs.uk/), MOD contract reference 700030303. The funders had no role in study design, data collection and analysis, decision to publish, or preparation of the manuscript.

**Competing interests:** I have read the journal's policy and the authors of this manuscript have the following competing interests: PT, LA and NK are contracted by the Healthcare Quality Improvement Partnership (HQIP) to deliver the Mental Health Clinical Outcome Review Programme (MHCORP); LA is the Chair of the National Suicide Prevention Strategy Advisory Group (England), Department of Health and Social Care; NK is a member of the National Suicide Prevention Strategy Advisory Group (England), Department of Health and Social Care and chaired the 2022 guideline development group for the National Institute for Health and Care Excellence (NICE) depression in adults' guidelines and was the a topic advisor for the 2022 National Institute for Health and Care Excellence (NICE) self-harm guidelines. Members of the study team (KH, and HD and AB) were are employees of the MoD and NHSE at the time of the study.

**Abbreviations:** CI, confidence interval; CTS, Complex Treatment Service; DCMH, Departments of Community Mental Health; HIS, High-Intensity Service; HQIP, Healthcare Quality Improvement Partnership; HR, hazard ratio; IAPT, Improving Access to Psychological Therapies; IQR, interquartile range; MPGS, military provost guard services; NCO, noncommissioned officer; NCISH, National Confidential Inquiry into Suicide and Safety in Mental Health; OR, odds ratio; PTSD, post-traumatic stress disorder; QNVMHS, Quality Network for Veterans Mental Health Services; RAF, Royal Air Force; SLD, Service Leavers Database; SMR, standardised mortality ratio; TILS, Transition, Intervention and Liaison Service; UK, United Kingdom; UKAF, UK Armed Forces; US, United States; VCHA, Veterans Covenant Healthcare Alliance.

youngest age groups (10% for 16– to 19–year–olds; 23% for 20– to 24–year–olds). Study limitations include the fact that information on veterans was obtained from administrative databases and the role of pre–service vulnerabilities and other factors that may have influenced later suicide risk could not be explored. In addition, information on contact with support services was only available for veterans in contact with specialist NHS mental health services and not for those in contact with other health and social care services.

## Conclusions

In this study, we found suicide risk in personnel leaving the UKAF was not high but there are important differences according to age, with higher risk in young men and women. We found a number of factors which elevated the risk of suicide but deployment was associated with lower risk. The focus should be on improving and maintaining access to mental health care and social support for young service leavers, as well as implementing general suicide prevention measures for all veterans regardless of age.

## Author summary

### Why was this study done?

- There are few international studies investigating suicide in veterans and no recent UK–wide studies—we previously published findings from a 10–year (1996 to 2005) study of suicide in almost 234,000 people who left the UK Armed Forces (UKAF).

- Since this publication, the nature of serving in the UK military has changed. Periods of intense activity in Iraq and Afghanistan have resulted in concern about the mental health impact of service. The number of personnel in the Army has decreased. There are also more services available for veterans experiencing poor mental health. Patterns of suicide in the general population have changed.

### What did the researchers do and find?

- We linked national databases of people who left the UKAF (collected by the Ministry of Defence) and people who died by suicide (collected by the National Confidential Inquiry into Suicide and Safety in Mental Health) to investigate the rate, timing, and risk factors for suicide in veterans.

- Between 1996 and 2018, 458,058 people left the UKAF and 1,086 (0.2%) died by suicide. The findings indicate that the overall suicide rate in veterans was similar to the general population (if anything, it was slightly lower), although the risk of suicide in men and women aged under 25 years who had left the military was 2 to 3 times higher than the same age group in the general population.

- Risk factors for suicide in veterans of the UKAF included being male, serving in the Army, being discharged from the military between the ages of 16 and 34 years, being untrained, and having a length of service of under 10 years. A quarter of veterans who died by suicide had been in contact with mental health services in the year before they died.

**What do these findings mean?**

- The findings from this study indicate that the overall risk of suicide in veterans is not high but that young men and women leaving the UKAF are at higher risk, especially if they have had a short length of service.

- Improving access to mental health care and social support for people who have left the UKAF, particularly if they are young, is important for suicide prevention in this group and may need to be maintained in the long–term. Given the number of older veterans who die by suicide, general suicide prevention measures are also needed for all veterans regardless of age.

- We were unable to explore the role of pre–service or other factors that may have influenced late suicide risk, such as childhood trauma, unemployment, or homelessness— these factors are important in this population and need further investigation.

## Introduction

Suicide accounts for around 700,000 deaths per year across the world [1] and its prevention is a global health priority [1]. Although suicide rates are generally lower in people serving in the Armed Forces than in the general population [2–4], data suggest the number of suicide deaths by male army personnel in the United Kingdom (UK) and United States (US) has increased in recent years [3,4]. Despite considerable policy interest in the mental health of veterans [5], research investigating suicide in this group is relatively sparse and often conflicting. In the US, suicide rates for veterans peaked in 2018 but have since fallen [6]. Studies in the US have also shown veterans to have a higher rate of suicide compared to the US general population [6–8]. Similar patterns have been reported in Australian and Canadian veterans, with higher suicide rates reported in female veterans, in particular, compared to the female general population [9,10]. In male veterans of the Canadian Armed Forces, suicide risk has been consistently higher than the Canadian general population since 1976, although it has fallen, albeit not significantly, in more recent years [10]. Retrospective cohort studies of UK, Dutch, and deployed Swedish veterans, however, report no overall increase in suicide risk compared with their respective general populations [11–14].

There is a lack of global consensus on whether past military service increases subsequent suicide risk, but individual studies have investigated factors that may be associated with veterans' suicide. Increased suicide risk has been reported in young (24 years and under) male veterans, older (over 40 years) female veterans, veterans with depression or alcohol problems, those subject to early discharge from the Armed Forces, and those who left service more than 20 years previously [11,12,15–18]. The literature also suggests that veterans of the Armed Forces may be a potentially vulnerable group because of adverse life events prior to enlisting, childhood trauma, the difficulties associated with the transition to civilian life, high rates of unemployment following discharge, homelessness, and alcohol and drug misuse [17–22]. Evidence on the link between deployment and suicide risk is mixed with some US studies reporting increased risk in those who were deployed [23,24] and others reporting reduced risk [7].

There have been few systematic investigations of suicide in UK veterans. We have previously published the findings of a 10-year retrospective study of suicide in a cohort of almost 234,000 individuals who had left the UK Armed Forces (UKAF) between 1996 and 2005 [12].

In this cohort, 224 (0.1%) veterans died by suicide. Although we found the overall rate of suicide was not greater than that in the general population, the risk of suicide in young male veterans was elevated. A more recent study of nearly 80,000 Scottish veterans also found no overall increase in suicide risk compared to the wider population, although the study did report an increased suicide risk in older female veterans with suicide most common in mid-life [11].

Since the publication of our previous study [12], the military context in the UK has changed. A period of intensive operational activity has ended (e.g., Iraq, Afghanistan), with related concern about the mental health impact of having served in these conflicts [25]. The number of full-time personnel in the Army has decreased (by 19% since 2012) [26] and the age distribution has also changed with fewer recruits aged under 18 years [27]. A range of new community-based NHS mental health services for veterans, collectively named "Op COURAGE," in conjunction with the creation of a specific body within the NHS with Armed Forces expertise [28], may have improved access to mental health care in England. The devolved administrations have their own mental health support provision for veterans, e.g., the Veterans First Point Service in Scotland. Third sector organisations also continue to raise awareness and deliver support and services to those who have left the UKAF [29,30]. In addition, the Quality Network for Veterans Mental Health Services (QNVMHS) [31], developed in collaboration with the Contact Group [32], aims to improve mental health services and care for veterans. The Government's Armed Forces Covenant, published in 2011 [33], has also pledged to treat those who serve or have served in the UKAF fairly and introduced the requirement within the NHS Constitution [34] that "veterans are not disadvantaged in accessing health services." Most recently, the Veterans' Strategy Action Plan [35] has detailed the steps the UK Government will take to better support the veteran community by 2028. There is potentially greater public awareness and reduced stigma associated with mental health problems in general [36]. In the general population, there has been a shift in the pattern of age-related suicide rates; men aged 45 to 64 years are the group at highest risk of suicide, where previously (before 2010) it was younger men aged 25 to 44 years [37].

The specific aims of this study were to: (i) investigate age-specific rates of suicide in veterans and compare these with rates in the general and serving populations; (ii) identify risk factors, characteristics, and service contacts among veterans; and (iii) describe trends in suicide rates in veterans and compare these with trends in the serving and general populations.

## Method

### Study design

In this retrospective cohort study of personnel who left the UKAF, our main outcome was death by suicide after leaving service. We linked national databases of discharged UKAF personnel and deaths by suicide. Comparisons were made with both the general population and serving personnel. We conducted a case control analysis on a subset of this cohort identified as being in contact with mental health services in the 12 months prior to death. This study follows the Strengthening the Reporting of Observational studies in Epidemiology (STROBE) guidelines (S1 STROBE Checklist).

### Study setting and individuals

This study covered England, Scotland, and Wales (due to lack of availability of suicide data from Northern Ireland). Individuals who had had left any of the 3 branches of the UKAF (Royal Navy (including Royal Marines), Army, and Royal Air Force (RAF)) between 1 January 1996 and 31 December 2018 were included. There was no restriction on length of military

service; we included anyone who left after their first day of basic training, in accordance with the UK definition of a veteran [33]. The focus of this study was regular UKAF personnel. Individuals who had only served as reserves were excluded from the study because their experiences of service would have differed from those of regular personnel and their characteristics may reflect the general population more closely than they did the serving population [38]. Suicide risk in reservists will be examined in future work. "Other" assignment types such as cadet forces and military provost guard services (MPGS) were also excluded. Regulars included any individual who had been a regular of the UKAF at any point in their service record regardless of their assignment status at last discharge.

## Armed forces databases

**Service leavers database.** The MOD has a database of all service personnel who have left the UKAF since the mid-1970s (circa 2.1 million records) compiled from the MoD's Joint Personnel Administration system and legacy pay systems (pre-2005), known as the Service Leavers Database (SLD). Defence Statistics, the team within the MoD responsible for compiling staffing surveys and health statistics, compiled an extract from the SLD on all personnel alive on the day of exit for the period 1 January 1996 to 31 December 2018. The starting year was chosen because of the availability of clinical and general population data from that point onwards. The initial extract included a limited number of core variables (name, date of birth, date of exit from Service) that were used for linkage purposes. A further extract of pseudo-anonymised variables was provided once database linkage was complete (see database linkage subsection below).

## Suicide databases

We used 2 databases held by the National Confidential Inquiry into Suicide and Safety in Mental Health (NCISH) to identify deaths by suicide during the study period (1996 to 2018).

**General population suicide database.** The NCISH general population suicide database included all suicide deaths in the UK. The database is collated from national mortality data on all people who died by suicide obtained from the Office for National Statistics (for deaths registered in England and Wales), National Records of Scotland (for deaths registered in Scotland), and the Northern Ireland Statistics and Research Agency (for deaths registered in Northern Ireland). For this study, data on deaths in Northern Ireland were not included due to restrictions in the disclosure of confidential person identifiable data for health and social care purposes [39]. Deaths assigned International Classification of Diseases, Tenth Revision (ICD-10) codes X60-X84 (intentional self-harm) or ICD-10 codes Y10-Y34 (excluding Y33.9), Y87.0, and Y87.2 (events of undetermined intent) were included in the study sample, as is standard for suicide research in the UK (since the majority of these undetermined deaths are likely to be suicide deaths) [40]. These deaths will collectively be referred to as suicide deaths throughout this paper. The variables in the NCISH general population database included age, sex, and method of suicide.

**Database of suicide deaths in people in recent (< = 12 month) contact with NHS mental health services.** The NCISH patient database is a detailed clinical database of patients who have been in contact with specialist NHS mental health services in the 12 months prior to their death. From national data on all people who died by suicide, mental health providers identified which individuals had contact with NHS mental health services in the 12 months before death. Clinical information was collected via a questionnaire completed by the senior professional responsible for the patient's care. During the study period, response rates for NCISH questionnaires were 95%. NCISH data collection methods are described in detail elsewhere [41].

## Database linkage

We linked the database of those who had left the UKAF (the SLD) with the NCISH general population suicide database using last name, first name(s) (where available), and date of birth. Of the 1,086 individuals we linked between the 2 databases, 89% ($N$ = 973) were exact matches using all variables, and 4% ($N$ = 41) were matched using date of birth, last name, and initials. A further 3,745 individuals matched on date of birth and last name, these were manually checked by members of the research team (SI, CR, JW, LB), equivocal matches were discussed, and 7% ($N$ = 72) were verified as veteran suicide deaths based on consensus. Once data linkage was complete, 17 additional pseudo-anonymised variables were obtained from the UKAF databases as described above. These variables included information on demographic characteristics (e.g., age, sex, ethnicity, marital status) and military service (e.g., assignment type (regular/reserve), rank, training indicator, date of entry, length of service, deployment on combat operations (including the Falklands Campaign (1982), Gulf-1 (1990 to 1991), Iraq (2003 to 2011), and Afghanistan (2002 to 2014)), and date and type of discharge (including voluntary and end of contract leavers and involuntary (administrative, disciplinary, and medical) exits). Data were over 95% complete for these variables. The resulting dataset was then linked to the NCISH patient database using a NCISH-derived unique identifier. Names, dates of birth and dates of death, and any unlinked data, were stripped from the database once linkage was complete.

## Statistical analysis

All our statistical analysis was performed using STATA 16.1 software for Windows. Simple descriptive statistics with numbers and proportions were used to examine the characteristics of former service personnel who died by suicide. Data on ethnicity were excluded due to a high proportion of missing data (82%). We set out our survival analysis data using the STSET command in STATA and we calculated crude suicide rates by time since leaving the UKAF with person-years at risk as the denominator using the STPTIME command. This meant our denominator (i.e., person-years at risk) accounted for an individual's age moving between age categories as the study progressed. Person-years at risk for each veteran were calculated from the date of being discharged from the UKAF to the date of suicide, or to the end of the study period (31 December 2018), if the individual did not die by suicide. To compare suicide rates to the UK general population, we calculated age-specific mortality ratios in 5-year age bands and an overall standardised mortality ratio (SMR). In the oldest age bands, different age categories were collapsed for some analysis because of small numbers. We also examined the relationship between suicide rates in veterans and their age at death using a locally weighted scatterplot smoothing (lowess) curve. Age-specific suicide rates in the UK general population for the period 1996 to 2018 were calculated using the number of suicides from the NCISH general population database over the study period as the numerator and annual published mid-year population estimates from ONS as the denominator [42]. To investigate the risk factors that were most associated with suicide by veterans after discharge, we used Cox proportional hazards models with age at death by suicide as the failure time and the end of the study period (31 December 2018) as the censor time. We examined risk by generating unadjusted hazard ratios (HRs) and 95% confidence intervals (CIs), and created Kaplan–Meier survival curves for the risk groups. The rate of contact with NHS mental health services in the 12 months prior to death was determined using the NCISH patient database. To compare the characteristics of veterans who had died by suicide within 12 months of mental health service contact with individuals in service contact in the general population who died by suicide, a matched case-control design was used. We selected up to 5 controls (general population) per case (former

service personnel) matched on age, sex, and year of death from the NCISH patient database. These 1,357 controls had no record of serving in the UKAF. Assuming a prevalence of a risk factor of 30% in the control group and α = 0.05, 5 controls per case gave us 80% power to examine relative risks of 2.5 and above. Conditional logistic regression analysis was used to explore differences between the cases and controls and unadjusted differences are presented using conditional odds ratios (ORs) and 95% CIs. To compare suicide rates in veterans with the rate in serving personnel, we calculated age-specific mortality ratios in 5-year age bands and an SMR using published data on suicide in serving personnel between 1996 and 2018 [3] as a comparator. We do not present age-specific rates for women, as published comparator data were unavailable due to small numbers. Cell counts under 5 (including zero) are not presented in accordance with guidance from the MoD on statistical disclosure control to safeguard confidentiality [43].

## Ethical approval

The study was approved by the Ministry of Defence Research Ethics Committee (2071/MOD-REC/21). Exemption under Section 251 of the NHS Act 2006, enabling access to confidential and identifiable information without informed consent in the interest of improving care, was obtained from the Health Research Authority Confidentiality Advisory Group (21/CAG/0050) and the Public Benefit and Privacy Panel for Health and Social Care (2021–0290). Data were provided by the Healthcare Quality Improvement Partnership (HQIP) from the Mental Health Clinical Outcome Review Programme (MH-CORP) delivered by the National Confidential Inquiry into Suicide and Safety (NCISH) (HQIP362).

## Results

### Characteristics of discharged veterans and those who died by suicide

We obtained data on a total of 458,048 veterans who served as regulars and had left the UKAF between 1 January 1996 and 31 December 2018. These individuals accumulated a total of 5,852,124 person years at risk and a median length of follow-up (interquartile range (IQR)) of 12.9 years (7 to 18 years). The cohort was predominantly male (416,254, 91%) and the median age (IQR) of the cohort at last discharge from the UKAF was 26 years (21 to 37 years). Overall, 283,111 (62%) had served in the Army, 88,664 (19%) in the Naval Service, and 86,273 (19%) in the RAF. Overall, 38,722 (8%) had a medical reason for discharge.

Based on linkage between NCISH and MoD databases, 1,086 (0.2%) individuals who had served as regulars were found to have died by suicide after leaving the UKAF. Their median age (IQR) at discharge was 23 years (19 to 31 years); their median age (IQR) at death was 32 years (26 to 42 years). The majority (1,046, 96%) were male. Around 19% ($n$ = 203) of the 1,086 veterans who died by suicide were aged under 25 years, 63% ($n$ = 682) were aged between 25 and 44 years, and 19% ($n$ = 201) were 45 years or older. Overall, 799 (74%) had served in the Army, 165 (15%) in the Naval Service, and 122 (11%) in the RAF. The most common method of suicide was hanging or strangulation ($n$ = 672, 62%), followed by self-poisoning ($n$ = 155, 14%). Firearm deaths were rare ($n$ = 27, 2%). Hanging or strangulation was more common in veterans than in the general population during this time period (62% versus 44%; $p$ < 0.001) and self-poisoning less common (14% versus 24%; $p$ < 0.001). Firearm deaths were similar to the general population (2%; $p$ = 0.31). A total of 502 (46%) of the 1,086 veterans who died by suicide were early service leavers (with less than 3 years of service). Of these 502 individuals, 165 (33%) were aged 24 years and younger at the time of suicide.

## Rate of suicide

Table 1 shows the age-specific mortality ratio and the SMR for men who left the UKAF compared to the general population. The risk of suicide for male veterans was similar to the risk of suicide in the age-matched general population. Overall, the SMR and 95% CIs suggested the veterans' suicide rate was very slightly (but statistically significantly) lower than the rate in the general population (SMR [95% CI] 94 [88 to 99]). However, the risk of suicide for the 2 youngest veteran age groups (16 to 19 and 20 to 24 years) was approximately 2 to 3 times higher than their counterparts in the general population (age-specific mortality ratio [95% CI] of 160 [136 to 187] in those aged 20 to 24 years and 305 [220 to 422] in those aged 16 to 19 years). For men aged over 35 years, the age-specific mortality ratios suggested the risk of suicide was lower than for age-matched groups in the general population. Overall, the suicide rate appeared to fall with age from 28.0 per 100,000 person years at ages 16 to 19 to 8.8 per 100,000 person years at ages 55 to 59 (Table 1, Fig 1).

We do not show the breakdown of age-specific data in female veterans due to small numbers. The overall SMR for women who left the UKAF indicated that their risk of suicide was not greater than the risk of suicide in the general population (SMR [95% CI] 126 [92 to 172]). However, similar to male veterans, there was an increased risk of suicide in the 2 youngest female age groups compared to the same groups in the general population (age-specific mortality ratios [95% CI] 409 [102 to 1,633] in women aged 16 to 19 years and 310 [161 to 596] in women aged 20 to 24 years). Unlike older male veterans, older female veterans were not at reduced risk but numbers were small.

Suicide rates across services indicated a slightly elevated SMR for men who had served in the Army only (SMR [95% CI] 111 [103 to 119]), with once again the highest risks in the

**Table 1. Numbers, crude rates per 100,000 person years and age–specific rate ratios for suicide in male regulars who left the UKAF, 1996–2018.**

| Age | Number of suicide deaths | Crude suicide rate in veterans | Age-specific rate-ratio[a] (95% CI) | SMR[b] (95% CI) |
|---|---|---|---|---|
| **16–19** | 36 | 28.0 | 304.5 (219.7–422.2) | - |
| **20–24** | 156 | 27.5 | 159.6 (136.4–186.7) | - |
| **25–29** | 198 | 22.2 | 111.0 (96.6–127.6) | - |
| **30–34** | 207 | 22.9 | 102.8 (89.7–117.8) | - |
| **35–39** | 129 | 18.4 | 80.1 (67.4–95.2) | - |
| **40–44** | 121 | 18.1 | 75.2 (62.9–89.9) | - |
| **45–49** | 90 | 15.6 | 66.6 (54.2–81.9) | - |
| **50–54** | 65 | 15.8 | 74.9 (58.7–95.5) | - |
| **55–59** | 25 | 8.8 | 47.2 (31.9–69.8) | - |
| **60–64** | 13 | 9.2 | 62.1 (36.0–106.9) | - |
| **65–87** | 6 | 6.1 | 49.5 (22.3–110.3) | - |
| **16–87[c]** | **1,046** | **19.5** | - | **93.5 (88.0–99.3)** |

[a]Using the general population as the reference population (Source: ONS, Population estimates time series dataset, June 2021 [42]. Age–specific rate–ratios are expressed so that a ratio under 100 indicates reduced risk and a ratio above 100 indicates an increased risk of suicide.

[b]An SMR indicates the overall risk of this cohort compared to the general population.

[c]Age category up to 87 because no suicide deaths in male veterans serving as regulars occurred in older age groups.

CI, confidence interval

SMR, standardised mortality ratio

UKAF, UK Armed Forces.

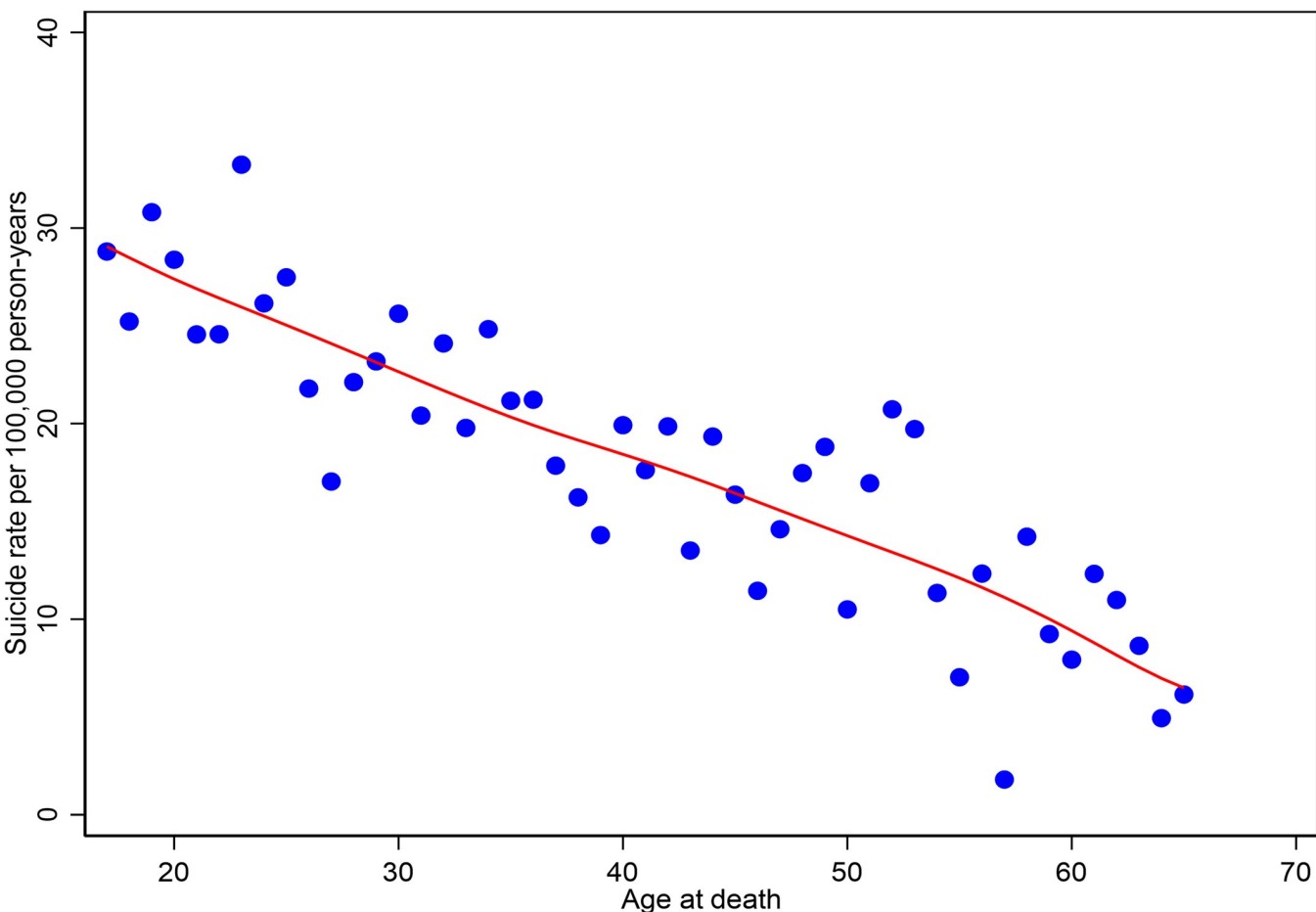

**Fig 1. Suicide rates in veterans by age at death (in single years), with locally weighted scatterplot smoothing (lowess) curve.**

youngest age groups (age-specific mortality ratios [95% CIs] for males aged 16 to 19 years: 331 [233 to 471], 20 to 24 years 182 [154 to 215], 25 to 29 years 128 [110 to 149], and 30 to 34 years 120 [103 to 139]). The risk of suicide for men who had served in the Naval Service or RAF was not greater than the risk of suicide in the age-matched general population (SMRs [95% CIs] 69 [59 to 81] and 61 [51 to 73], respectively).

## Timing of suicide

Fig 2 shows the rate of suicide by time elapsed since discharge from the UKAF. These data are presented for men only because of the small number of women in individual time period categories. There was some year-on-year variation, but the risk of suicide appeared to be persistent.

## Risk factors for suicide

Higher suicide risk was associated with male sex, being discharged from the UKAF between the ages of 16 and 34 years, being untrained on discharge, having served for less than 10 years, and receiving an administrative, disciplinary, or medical discharge (Table 2). Being married, being a commissioned officer, being an NCO (noncommissioned officer), and deployment on combat operations were associated with lower suicide risk (Table 2). Serving in the Navy and

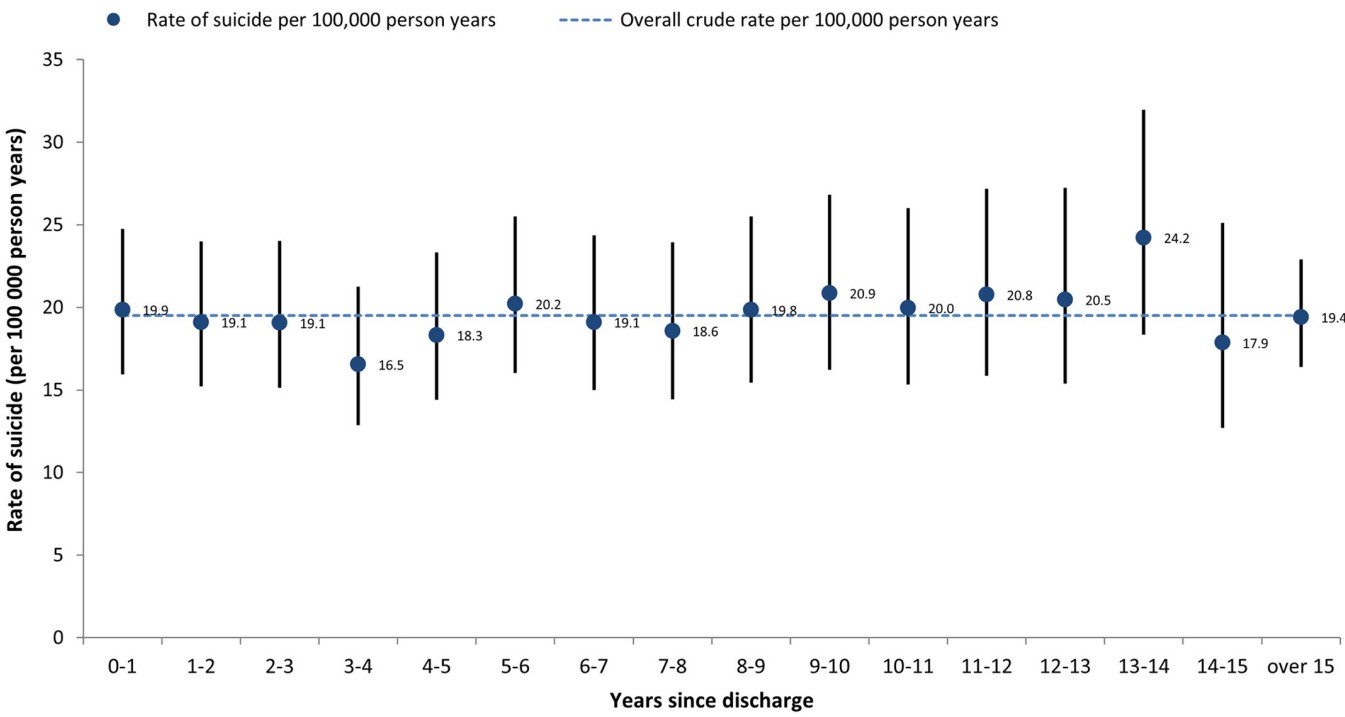

**Fig 2. Rate of suicide in male veterans serving as regulars by time elapsed since discharge.**

the RAF were also associated with lower suicide risk compared to serving in the Army. Kaplan–Meier survival estimates of veterans who died by suicide according to selected risk factors are presented in S1 Fig. The curves are consistent with the HRs presented in Table 2.

## Rates and characteristics of veterans in contact with mental health services prior to suicide

Of the 1,086 individuals who died by suicide after leaving the UKAF, 273 (25%, 95% CI 22 to 28) had been in contact with NHS mental health services in the 12 months prior to their death. This is similar to the proportion of individuals in contact with NHS mental health services in the 12 months before suicide in the general population (27%). The proportion of individuals who had left the UKAF and were mental health patients prior to suicide was lowest in the youngest age groups (10% in those aged 16 to 19 years; 23% in those aged 20 to 24 years; compared to 18% of those aged 16 to 19 years and 23% of those aged 20 to 24 years in recent (12 month) contact with NHS mental health services in the general population). It should be noted that these proportions should not be interpreted as suggesting young veterans were less likely to have contact with mental health services than young people in the general population because the numbers in the youngest age categories were very small.

Table 3 compares the characteristics of veterans who died by suicide within 12 months of contact with NHS mental health services (cases) with individuals matched by age, sex, and year of death, who also died by suicide within 12 months of contact with NHS mental health services but had not served in the UKAF (controls). There were few differences between cases and controls. Veterans of the UKAF who died by suicide were more likely to have a primary diagnosis of affective disorder (depression or bipolar disorder), evidence of a post-traumatic stress disorder (PTSD) diagnosis (including a primary, secondary, tertiary, or quaternary diagnosis, although numbers were small), and to have had their first contact with NHS mental

**Table 2. Risk factors for suicide.**

| Variable | Categories | N (at risk)[a] | Number dying by suicide | HR[b] (95% CI) | p-Value[c] |
|---|---|---|---|---|---|
| **Age (years) at service exit** | 16–19 | 75,503 | 287 | 2.62 (1.97–3.49) | |
| | 20–24 | 114,562 | 321 | 2.25 (1.69–2.98) | |
| | 25–29 | 87,914 | 180 | 1.75 (1.30–2.37) | |
| | 30–34 | 51,354 | 93 | 1.43 (1.03–1.99) | |
| | 35–39 | 38,602 | 70 | 1.20 (0.84–1.71) | |
| | 40–44 | 49,662 | 79 | 1.18 (0.84–1.67) | |
| | 45+[d] | 40,451 | 56 | Base | <0.001 |
| **Sex** | Male | 416,254 | 1,046 | Base | <0.001 |
| | Female | 41,794 | 40 | 0.37 (0.27–0.51) | |
| **Service** | Army | 283,111 | 799 | Base | <0.001 |
| | Navy | 88,664 | 165 | 0.61 (0.51–0.72) | |
| | RAF | 86,273 | 122 | 0.52 (0.43–0.63) | |
| **Rank** | Non-officer rank[e] | 252,299 | 761 | Base | <0.001 |
| | Commissioned Officer | 44,659 | 41 | 0.28 (0.20–0.38) | |
| | NCO | 160,675 | 271 | 0.51 (0.44–0.58) | |
| **Marital status** | Single | 248,231 | 727 | Base | <0.001 |
| | Married | 139,918 | 227 | 0.51 (0.44–0.59) | |
| | Divorced/separated | 7,681 | 10 | 0.72 (0.39–1.35) | |
| **Training[f]** | Trained | 270,112 | 579 | Base | <0.001 |
| | Untrained | 141,363 | 404 | 1.27 (1.12–1.44) | |
| **Type of discharge** | Normal | 82,215 | 134 | Base | <0.001 |
| | Disciplinary | 26,252 | 51 | 2.67 (1.93–3.69) | |
| | Medical | 38,722 | 122 | 2.55 (1.99–3.26) | |
| | Administrative | 86,924 | 357 | 2.88 (2.36–3.52) | |
| | Own request | 107,320 | 191 | 1.36 (1.09–1.69) | |
| **Deployed on combat operation[g]** | No | 328,875 | 899 | Base | <0.001 |
| | Yes | 129,173 | 187 | 0.71 (0.60–0.83) | |
| **Length of service (years)** | <1 | 96,653 | 328 | 2.18 (1.82–2.61) | |
| | 1 to <2 | 22,790 | 98 | 3.11 (2.43–3.97) | |
| | 2 to <3 | 16,230 | 76 | 3.41 (2.61–4.45) | |
| | 3 to <4 | 21,798 | 60 | 1.87 (1.40–2.50) | |
| | 4 to <5 | 35,705 | 71 | 1.59 (1.21–2.09) | |
| | 5 to <10 | 104,455 | 203 | 1.51 (1.23–1.84) | |
| | 10 to <15 | 45,245 | 65 | 0.96 (0.72–1.27) | |
| | 15+ | 115,172 | 185 | Base | <0.001 |

[a]Numbers for each category may not tally with the total number at risk due to missing data.

[b]Unadjusted HRs calculated using Cox proportional hazards modelling. Males and females both included. Total number at risk 458,048, total number of suicide deaths 1,086.

[c]p–Values calculated using log–rank test.

[d]Older age categories collapsed because of small numbers of individuals who left service in older age groups.

[e]Non–officer rank refers to individuals with the rank of able rate, marine, private, aircraftman, lance corporal, or senior aircraftman.

[f]Trained refers to individuals who have completed Phase 1 (basic) and Phase 2 (skill–based) training.

[g]Includes deployment to Falklands Campaign (1982), Gulf–1 (1990–1991), Iraq (2003–2011), and Afghanistan (2002–2014).

CI, confidence interval; HR, hazard ratio; NCO, noncommissioned officer.

**Table 3. Characteristics of individuals who left the UKAF and who had contact with mental health services in the 12 months prior to suicide and matched controls who had not served in the UKAF.**

| Risk group | Variable | Cases (n = 273)[a] | Controls (n = 1,357)[a] | p-Value | OR (95% CI) |
|---|---|---|---|---|---|
| **Demographic features** | Unmarried | 200 (78%) | 1,053 (81%) | 0.211 | 0.81 (0.58–1.12) |
| | Unemployed | 156 (61%) | 758 (61%) | 0.998 | 1.00 (0.76–1.32) |
| | Long-term sick | 19 (7%) | 157 (13%) | 0.022 | 0.57 (0.35–0.95) |
| | Living alone | 121 (48%) | 602 (48%) | 0.820 | 1.03 (0.78–1.36) |
| | Homeless | 10 (4%) | 60 (5%) | 0.721 | 0.88 (0.44–1.77) |
| **Service characteristics** | In-patient | 21 (8%) | 108 (8%) | 0.844 | 0.95 (0.58–1.55) |
| | Recent (<3 months) discharge | 38 (15%) | 193 (16%) | 0.911 | 0.98 (0.66–1.44) |
| | Under CR/HT services | 30 (12%) | 114 (9%) | 0.146 | 1.39 (0.90–2.15) |
| | Missed last contact | 73 (30%) | 357 (30%) | 0.689 | 1.06 (0.78–1.44) |
| | Non-adherent with medication | 29 (13%) | 173 (15%) | 0.642 | 0.90 (0.59–1.39) |
| **Clinical characteristics** | Schizophrenia and other delusional disorders | 41 (15%) | 325 (25%) | 0.0006 | 0.55 (0.39–0.79) |
| | Affective disorders (depression or bipolar disorder) | 90 (33%) | 363 (27%) | 0.041 | 1.35 (1.01–1.80) |
| | Alcohol dependence/misuse | 31 (11%) | 129 (10%) | 0.375 | 1.21 (0.80–1.84) |
| | Drug dependence/misuse | 24 (9%) | 147 (11%) | 0.266 | 0.77 (0.49–1.23) |
| | Personality disorder | 34 (13%) | 122 (9%) | 0.101 | 1.43 (0.94–2.16) |
| | Any PTSD diagnosis[b] | 9 (3%) | 14 (1%) | 0.011 | 3.29 (1.37–7.87) |
| | Any secondary diagnosis | 154 (57%) | 784 (59%) | 0.557 | 0.92 (0.70–1.21) |
| **Behavioural characteristics** | History of illness <12 months | 59 (24%) | 226 (19%) | 0.095 | 1.33 (0.96–1.85) |
| | History of self-harm | 173 (68%) | 831 (64%) | 0.290 | 1.17 (0.87–1.57) |
| | History of violence | 77 (32%) | 392 (32%) | 0.954 | 0.99 (0.73–1.35) |
| | History of alcohol misuse | 160 (62%) | 735 (57%) | 0.125 | 1.24 (0.94–1.63) |
| | History of drug misuse | 133 (52%) | 745 (57%) | 0.145 | 0.80 (0.60–1.08) |
| **Contact with services** | First contact with MHS <12 months | 89 (36%) | 340 (28%) | 0.015 | 1.45 (1.08–1.95) |
| | Last contact <7 days | 101 (38%) | 569 (43%) | 0.133 | 0.81 (0.62–1.07) |
| | Long-term risk: low or none | 134 (60%) | 656 (58%) | 0.671 | 1.07 (0.79–1.43) |
| | Short-term risk: low or none | 190 (81%) | 1,016 (84%) | 0.251 | 0.80 (0.56–1.16) |

[a]Percentages in brackets expressed as a proportion of valid cases. The number of valid cases may vary between variables (risk groups) due to missing data.

[b]Any PTSD diagnosis includes a primary, secondary, tertiary, or quaternary diagnosis of PTSD.

CI, confidence interval; CR/HT, Crisis Resolution/Home Treatment; MHS, Mental Health Services; OR, odds ratio; PTSD, post–traumatic stress disorder; UKAF, UK Armed Forces.

health services within the 12 months prior to death. They were less likely to have been on long-term sick leave at the time of death and to have a diagnosis of schizophrenia and other delusional disorders.

## Comparison with the in-service sample

Table 4 presents the rates of suicide for men who left the UKAF and men who were serving in the UKAF during the period 1996 to 2018. The age-specific mortality ratios and SMR compare rates of suicide in men who left the UKAF to rates of suicide in male serving personnel. The results suggested the overall risk of suicide was over twice as high in men discharged from the UKAF than those still serving (SMR [95% CI] 228 [214 to 242]), with elevated risks across all age groups. Published age-specific suicide rates for serving women were not available to calculate an SMR.

**Table 4. Number and crude rates per 100,000 person years and age–specific rate ratios for suicide in male regulars who left the UKAF and male regulars serving in the UKAF at the time of death, 1996–2018*.**

| Age | Discharged sample | | In-service sample | Age-specific rate ratio (95% CI)[b] | SMR (95% CI)[c] |
|---|---|---|---|---|---|
| | Number of suicide deaths | Crude suicide rate in veterans | Average age-standardised suicide rate[a] | | |
| **16–19** | 36 | 28.0 | 13.9 | 201.7 (145.5–279.7) | - |
| **20–24** | 156 | 27.5 | 11.8 | 233.3 (199.4–272.9) | - |
| **25–29** | 198 | 22.2 | 9.6 | 231.5 (201.4–266.1) | - |
| **30–34** | 207 | 22.9 | 7.9 | 290.4 (253.4–332.8) | - |
| **35–39** | 129 | 18.4 | 7.8 | 236.4 (198.9–280.9) | - |
| **40–44** | 121 | 18.1 | 10.9 | 165.6 (138.6–198) | - |
| **45–54** | 155 | 15.7 | 7.2 | 217.7 (186–254.8) | - |
| **16–54**[d] | **1,002** | **22.8** | **9.2** | - | **227.6 (213.9–242.1)** |

*Numbers differ to Table 1 because of different age categories.

[a]Source: Ministry of Defence, UK Armed Forces suicides: 2021 [3].

[b]Using the in–service population as the reference population. Age–specific rate–ratios are expressed so that a ratio under 100 indicates reduced risk and a ratio above 100 indicates an increased risk of suicide, and compare suicide rates in the discharged (veteran) sample with rates in the serving sample.

[c]An SMR indicates the overall risk of this cohort compared to the serving population.

[d]Age category restricted to 54 years because no in–service suicide deaths in serving male regulars occurred in older age groups.

CI, confidence interval; SMR, standardised mortality ratio; UKAF, UK Armed Forces.

## Discussion

### Main findings

We found that overall, veterans were not at an increased risk of suicide when compared to the general population. However, the suicide risk was 2 to 3 times higher in men and women aged under 25 years who had left the UKAF. The risk was lower in veterans aged 35 years and older compared to the general population. We found some year-on-year variation in the suicide rate in the years after discharge from the UKAF but the risk of suicide was fairly persistent. Men, those who had served in the Army, and those with a length of service of less than 10 years were at greater risk of suicide. Other factors associated with a higher risk of suicide were being discharged from services between the ages of 16 and 34 years, being untrained on discharge, and leaving service involuntarily (i.e., receiving an administrative, disciplinary, or medical discharge). We found higher rank, being married, and deployment on combat operations were associated with lower suicide risk. Methods of suicide in veterans were comparable to the general population, although hanging and strangulation were more common (62% versus 44%) and self-poisoning less common (14% versus 24%) in veterans—potentially reflecting the sex composition of the veteran cohort compared with the general population.

A quarter (25%) of veterans who died by suicide had been in contact with specialist NHS mental health services in the 12 months prior to death (i.e., were patient suicides). The rate of contact with these services was lowest in the age group at the greatest risk of suicide (21% for those aged 24 years or younger). A primary diagnosis of affective disorder was more common and a diagnosis of schizophrenia less common in veterans who were mental health patients (i.e., cases) compared to mental health patients who had not served in the UKAF (i.e., controls). Evidence of a PTSD diagnosis was uncommon but more likely in cases than controls (3% versus 1%), although CIs were wide (OR 3.29 (1.37 to 7.87)) given the small number of veterans (9) with this diagnosis. There were few other differences between our cases and controls. Our examination of suicide rates in veterans compared with serving personnel indicated that suicide risk was over 2 times higher in veterans, with elevated risks across all age groups.

The findings from this current study are broadly consistent with our previous study [12] and with others examining suicide in UK [11] and US veterans [18].

## Strengths and limitations

This study is one of the few to investigate suicide in military veterans and includes a large cohort of individuals to have left the UKAF, with up to 23 years follow-up and linkage to multiple databases. Using a whole population approach, it includes over 200,000 additional veterans and an additional 13 years of data compared to our previous study of suicide in UK veterans [12]. This has enabled a more comprehensive and contemporaneous examination of the burden of suicide in UK veterans and allowed us to identify key characteristics that could inform preventive efforts. Our findings, however, need to be considered in the context of several limitations.

Information on those who had left the UKAF was obtained from administrative databases and we were limited by the information contained in those. We were unable to explore the role of pre-service vulnerabilities or factors that may have influenced later suicide risk (for example, unemployment and homelessness). These factors will be examined in the next phase of this study where we will undertake an in-depth examination of a sample of veteran suicide deaths using information from coroner records.

In the absence of a comprehensive national database, it was not feasible for us to include a matched comparison group of nonveterans for the main analysis. In that sense, we were not comparing "like with like" with respect to suicide risk in Table 1, but that was not our aim—we sought to compare suicide rates in veterans to the general population as a whole. A case control study would have necessitated a general population control group on who equivalent data were available. However, we do acknowledge that other studies have used this methodology [11].

Information on contact with support services was only available for those veterans identified as having been in contact with specialist NHS mental health services. This study could not tell us the number of veterans who had been in contact with their GP prior to suicide, attended the Emergency Department, or who had been in contact with other health, social care, or voluntary services including military Departments of Community Mental Health (DCMH). We aim to examine contact with a range of support services in the next phase of this study. Our analysis of clinical characteristics was also based on a help-seeking sample of veterans in recent contact with specialist NHS mental health services (a quarter of the total) who may not have been typical of veterans as a whole and may have differed from veterans seeking help from other sources.

We examined suicide and undetermined deaths only. We did not examine other categories of death which may sometimes include possible suicide deaths (e.g., accidental deaths) or investigate wider causes (e.g., drugs and alcohol, natural causes). Such deaths may be important contributors to early mortality, although evidence suggests both Falkland and Gulf veterans have similar rates of death for external causes of injury and lower rates of death from disease than the general UK population, attributed to the "healthy worker effect" [44]. Although an additional 13 years of data were included compared to our previous study [12], this is shorter than some other studies examining veteran suicides [11] and limits the number of individual deaths we were able to include in the analysis. In the oldest age categories (55 to 87 years) particularly there were fewer than an average of 2 suicide deaths per year over the study period and this may have impacted on the power of our study and the statistical significance of some of our findings. Furthermore, veterans were followed-up for a maximum of 23 years for those who left the UKAF at the start of the study period (1996), less for those who left

later. With a median age at last discharge from the UKAF of 26 years, many of the veterans we examined would not have reached middle age by the end of the study period and those who did may not have been representative of all veterans in this age group. This means that our estimates of risk for older age groups may be less precise than our estimates for younger age groups.

In our analysis, we calculated SMRs using veterans suicide rates (based on a person-years denominator) with a general population rate (based on the aggregated number of suicide deaths from the NCISH general population database and UK population estimates) [42]. Although this is a standard epidemiological approach [45], it did not take into account levels of deprivation or consider geographical variations in suicide rates [37]. It is difficult to be certain whether or how this may have impacted our findings—basing calculations on general population rates in the North-East of England (which has a high general population suicide rate) would have led to lower SMRs for the veterans' population than using London rates (which have lower general population suicide rates). One approach might be to carry out analysis stratified by geographical area. We will be exploring spatial distribution of veterans' suicide deaths in future work, which will contextualise our findings on whether suicide risk in younger veterans is associated with the areas in which they live following discharge from the UKAF.

Veterans who had served only as reserves were excluded from our analysis. Reservists may not be exposed to the same experiences as regular personnel, although there is some evidence that deployed reservists may have poorer mental health outcomes than deployed regulars upon returning home from combat operations [46]. We plan to examine suicide risk in those who served only as reserves in a subsequent publication.

### Interpretation of findings

We found considerable overlap between several of the factors related to suicide in veterans of the UKAF, including younger age at discharge, short length of service, and being untrained. Such consistency in our findings suggests that young people who serve for only a short length of time represent a particularly vulnerable group. It seems plausible that this may partly reflect a selection effect—these young veterans were more prone to pre-service vulnerabilities rather than in-service or post discharge exposures that affected their ability to function for a longer period in the UKAF. Overall, we found younger veterans aged under 25 years were at higher risk of suicide than their counterparts in the general population. Although this finding is consistent with our previous study [12] and others examining suicide in US veterans [18], it differs from a recent Scottish study of military veterans [11], which identified a different at-risk group (men and women in middle age). This is likely due to differences in study design and setting, inclusion criteria, and time period. Regardless of the group at highest statistical risk, the number of older veterans who died by suicide in our study suggests that they are also an important part of the prevention challenge.

Some previous studies, but by no means all, have suggested a history of deployment is associated with higher suicide risk [23,24]. For some people, their experience of military service is a positive one but for others it may be negative. Public perception may be one of individuals traumatised by deployment on combat operations, struggling to settle back into society post-discharge, and not receiving the mental health care they need. While this might be the case for individual veterans, our results could be interpreted as presenting a somewhat different picture overall. Veterans who were younger and had lengths of service of less than 10 years were at greatest risk of suicide after discharge. Being older and longer lengths of service were protective, although it is possible that older veterans were more likely to be officers or NCOs (higher

rank was also protective) and rank (and by implication education and socioeconomic status) may be a factor in explaining the lower risk of suicide found in older veterans. A significant proportion of veterans had deployed on combat operations but there was no evidence that such deployment (and perhaps, by extension, combat-related exposure) was associated with suicide. In fact, deployment appeared to reduce risk. This finding could partly reflect what might be conceptualised as a "double healthy worker effect" with respect to suicide mortality—individuals with physical or psychological health problems will be less likely to be accepted into military service and may be less likely to be deployed on combat operations once in service. Although being selected for deployment does not routinely involve additional testing, postings selected for, or role within units, it could mean that more vulnerable individuals are less likely to be deployed or may already have left the UKAF. If this was the case, then those eligible for deployment would be healthier than those not deployed and healthier than those in the general population [14,47].

Other protective factors included being married and having completed training, perhaps highlighting the importance of social support and integration. However, we acknowledge the beneficial effect of being married could be diminished in those individuals whose marital status changed between discharge and death (as information on marital status was obtained from MOD administrative systems related to time in service). Individuals who had served in the Army and who had left service involuntarily were at higher risk of suicide. A minority of veterans, particularly younger veterans, were in contact with NHS mental health services, although this is true of men generally. A known diagnosis of PTSD was unusual, even in those under mental health care, but of course this was based on clinical assessments and people who did not present to services or patients where relevant symptoms were not recognised would not have been included. Interestingly, among patients who had sought mental health care, veterans had similar characteristics to nonveterans.

This study was unable to examine in detail the effect of pre- or post-service vulnerabilities or in-service exposures on suicide risk, or to estimate the relative contribution of each. Unemployment, financial and relationship problems, and pre-service adversity have been reported as common in veterans [19,22], but this is also true in the wider population, particularly among middle-aged men [48]. We plan to examine all potentially important antecedents in the next phase of this study using additional data sources such as coroners' records.

## Implications

Almost 1 in 25 people aged 16 years and over in England and Wales is a veteran of the UKAF [49]. The highest risk of suicide continues to be in young people who have left the UKAF and younger individuals with short lengths of service may have the most pressing needs. Those who have served in the Army and who leave service on a nonvoluntary basis are also at higher risk. However, prevention needs to go further than this. In terms of numbers, 16 to 24 year olds formed 18% of the total veteran population who died by suicide; older age groups (aged 40 to 54 years) made up 26% (even though their actual rates were lower). This suggests that all veterans regardless of age should be the focus of prevention.

Since the publication of our earlier study [12] over a decade ago, there have been considerable improvements to the care available for serving personnel and veterans, including, in the NHS, the introduction of specialist veteran-specific mental health services across the UK (e.g., the Transition, Intervention and Liaison Service (TILS), Complex Treatment Service (CTS), and High-Intensity Service (HIS)). The QNVMHS also provides accreditation to NHS and third sector mental health providers that meet agreed quality standards for military mental healthcare (31). The stigma surrounding mental health has also potentially reduced. The

availability of care and awareness of veterans' mental health needs within healthcare services, however, varies, and there remains a lack of knowledge regarding the Armed Forces Covenant within the healthcare system [5]. While it is known that support services for veterans may have increased in the UK in the last decade, the efficacy of that support requires further evaluation. The 2021 Census in England and Wales was also the first to ask if people had ever served in the UKAF [49]. As further data from this becomes available, a better understanding of the needs of veterans for service providers will develop.

The low rate of contact with specialist NHS mental health services in our study suggests veterans, particularly younger veterans, may not be seeking help. Although this mirrors findings for men in the general population, specific factors in veterans might be related to stigma or a perception that civilians may not understand the issues they are facing or have faced [50]. Evidence suggests it takes an average of 2 to 4 years for veterans to seek mental health support [5,51]. Interventions that encourage help-seeking or campaigns to reduce stigma lack evaluation but may increase engagement in this population. Equipping health services with the knowledge and training to better support the health of veterans and understand their culture and needs, such as through the Veteran Friendly GP Practice Accreditation Programme [52], the Veteran Aware Accreditation for all NHS Trust providers in England overseen by the Veterans Covenant Healthcare Alliance (VCHA) [53], and the Improving Access to Psychological Therapies (IAPT) positive practice guide for working with veterans [54], may continue to improve health outcomes and reduce suicide risk in, particularly younger, veterans.

A quarter of veterans who died by suicide were in contact with NHS mental health services in the 12 months prior to death and this is roughly equivalent to the proportion of suicide deaths by patients in the UK overall [55]. The characteristics of veteran and nonveteran patient suicides were comparable, suggesting the prevention challenge is similar in veterans and the general population, so tackling previous self-harm and alcohol or drug misuse, increasing awareness of the vulnerability of patients who live alone, and enhancing social support could be helpful strategies in veterans as they are in the general population. The persistence of suicide risk suggests that prevention may need to take a long-term perspective.

## Supporting information

**S1 STROBE Checklist. Strengthening the Reporting of Observational Studies in Epidemiology (STROBE) checklist.**
(DOCX)

**S1 Fig.** Kaplan–Meier survival estimates of veterans by: (a) age at service exit; (b) trained versus untrained; (c) deployed versus not deployed; (d) length of service in the UKAF.
(TIFF)

## Acknowledgments

We would like to thank Defence Statistics Health, the Armed Forces Team within NHS England, and staff at the National Confidential Inquiry into Suicide and Safety in Mental Health (NCISH) for their help and advice on the study. We thank Dan Stears, Fiona Naylor, and Liz Monaghan (members of Mutual Support for Mental Health Research (MS4MH-R), the patient and public involvement, and engagement group at the Centre for Mental Health and Safety, University of Manchester) and Tom Fox, Jo Brettell, and Wayne Palmer for their advisory roles in the study design. We would also like to thank the HQIP for the provision of data from the Mental Health Clinical Outcome Review Programme (MH-CORP) as delivered by the National Confidential Inquiry into Suicide and Safety in Mental Health.

## Author Contributions

**Conceptualization:** Cathryn Rodway, Pauline Turnbull, Andy Bacon, Kate Harrison, Nav Kapur.

**Formal analysis:** Cathryn Rodway, Saied Ibrahim, Jodie Westhead, Lana Bojanić, Harriet Dale, Nav Kapur.

**Funding acquisition:** Andy Bacon, Kate Harrison, Nav Kapur.

**Investigation:** Cathryn Rodway, Saied Ibrahim, Jodie Westhead, Lana Bojanić, Louis Appleby, Andy Bacon, Harriet Dale, Kate Harrison, Nav Kapur.

**Methodology:** Cathryn Rodway, Saied Ibrahim, Jodie Westhead, Lana Bojanić, Andy Bacon, Harriet Dale, Kate Harrison, Nav Kapur.

**Project administration:** Cathryn Rodway, Saied Ibrahim, Jodie Westhead, Lana Bojanić, Pauline Turnbull, Harriet Dale, Kate Harrison, Nav Kapur.

**Resources:** Pauline Turnbull, Louis Appleby, Andy Bacon, Harriet Dale, Kate Harrison, Nav Kapur.

**Supervision:** Cathryn Rodway, Saied Ibrahim, Lana Bojanić, Pauline Turnbull, Louis Appleby, Andy Bacon, Kate Harrison, Nav Kapur.

**Validation:** Jodie Westhead.

**Writing – original draft:** Cathryn Rodway, Saied Ibrahim, Nav Kapur.

**Writing – review & editing:** Cathryn Rodway, Saied Ibrahim, Jodie Westhead, Lana Bojanić, Pauline Turnbull, Louis Appleby, Andy Bacon, Harriet Dale, Kate Harrison, Nav Kapur.

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
