## [Editor Report · Decision Letter 0]

29 Jan 2023

Dear Dr Rodway, 

Thank you for submitting your manuscript entitled "Suicide after leaving the UK Armed Forces 1996-2018: a cohort study" for consideration by PLOS Medicine.

Your manuscript has now been evaluated by the PLOS Medicine editorial staff and I am writing to let you know that we would like to send your submission for external assessment.

However, before we can send your manuscript for full assessment, we need you to complete your submission by providing the metadata that is required. To this end, please login to Editorial Manager where you will find the paper in the 'Submissions Needing Revisions' folder on your homepage. Please click 'Revise Submission' from the Action Links and complete all additional questions in the submission questionnaire.

Please re-submit your manuscript within two working days, i.e. by Jan 31 2023 11:59PM.

Once your full submission is complete, your paper will undergo a series of checks in preparation for external assessment. 

Sincerely,

Richard Turner PhD

Consulting Editor, PLOS Medicine

plosmedicine@plos.org

---

## [Decision Letter · Decision Letter 1]

17 Mar 2023

Dear Dr. Rodway,

Thank you very much for submitting your manuscript "Suicide after leaving the UK Armed Forces 1996-2018: a cohort study" (PMEDICINE-D-22-04050R1) for consideration at PLOS Medicine. 

Your paper was discussed among the editors and sent to independent reviewers, including a statistical reviewer. The reviews are appended at the bottom of this email and any accompanying reviewer attachments can be seen via the link below:

[LINK]

In light of these reviews, we will not be able to accept the manuscript for publication in the journal in its current form, but we would like to invite you to submit a revised version that addresses the reviewers' and editors' comments fully. You will appreciate that we cannot make a decision about publication until we have seen the revised manuscript and your response, and we expect to seek re-review by one or more of the reviewers. 

We hope to receive your revised manuscript by Apr 07 2023 11:59PM. Please email us (plosmedicine@plos.org) if you have any questions or concerns.

Please let me know if you have any questions, and we look forward to receiving your revised manuscript. 

Sincerely,

Richard Turner PhD

Consulting editor, PLOS Medicine

plosmedicine@plos.org

We trust you will include additional details of the calculations, and if needed additional analyses, to respond to referee 1's comments.

Please add a new final sentence to the 'Methods and findings' subsection of the abstract which should begin "Study limitations include ..." or similar and should quote 2-3 of the study's main limitations. 

At line 36 (abstract), please consider adding "Study limitations include ..." or similar.

Please add an accessible 'Author summary' section after the Abstract, in non-identical prose. You may find it helpful to consult one or two recent research articles in PLOS Medicine to get a sense of the preferred style. 

Please trim the final paragraph of the Introduction (main text) to remove elements of discussion. Stating the aim of the study will suffice here. 

Please remove information on study funding from the end of the main text. In the event of publication, this information will appear in the article metadata via entries in the submission form. 

In the reference list, please convert all italics to plain text. 

Noting references 21 and 22, please list 6 author names consistently followed, where appropriate, with "et al.".

Thank you for including the completed STROBE checklist. Please rename the attachment 'S1_STROBE_Checklist" or similar and refer to it in the Methods section (Main text). 

In the checklist, please refer to items by section (e.g., 'Methods') and paragraph number, not by page numbers. 

Comments from the reviewers:

*** Reviewer #1: 

Thank you for the opportunity to review the revised version of this extremely important paper, a data linkage study of veteran suicides over the period 1 Jan 1996-31 Dec 2018. It is likely that this paper will inform both Government and Defence policy on suicide prevention, and therefore it is crucial that the findings are both valid and robust.

The authors had access to two large national databases, one recording Service leavers and one recording suicides and open verdicts nationally, and performed a data linkage to identify probable suicides among people with previous military service. A total of 1,086 suicides (0.2%) were identified among the 458,048 veterans who entered the database between 1 January 1996 and 31 December 2018. This is stated to cover a 22-year period (lines 21 and 131); I calculate it to cover 23 years.

I have considerable concerns with regard to the statistical analysis which is presented in the Results, particularly in lines 238-249. Table 1, showing a steady decline in "crude rate" from the youngest at death (16-19 years of age) to the oldest (45-54 years of age) is at marked variance with the nationally-reported age-specific rates for suicide, which show a near-normal distribution centred around middle age. It is difficult to understand how it could be plausible that a short period of military service (the median age at discharge for those who later died was 23 years) would so radically change the overall pattern of age at suicide. 

National data are based on annual prevalence rates per 100,000 population, but this becomes problematic when looking at veterans as the numbers in each age category are relatively small, eg a mean of less than 2 cases per year for the category aged 16-19 years at death as reported in Table 1. The authors have therefore used a rate of "per 100,000 person-years at risk". However they have not described how this was derived in the Methods, and I presume that it represents elapsed time between discharge from the military and death. The flaw in this method is that the longer the time between discharge and death (ie generally, the older the individual is at death), the more person-years they have been "at risk", and therefore the larger the denominator. This inevitably results in the declining "crude rate" seen in the third column of Table 1, but it is an artefact of the methodology, not a reduction in risk in older veterans. This has significant implications for policy. These observations are also relevant to Table 4.

The fourth column of Table 1 presents age-specific mortality ratios, using the general population as the reference population. The rates for the veterans are presumably those presented in column 3; the derivation of the corresponding rates for the population is not described but presumably comes from published national data. Are the population rates averaged over 23 years as for the veterans, or has a single year's rates been used and if so, which year? Furthermore, and importantly, the validity of calculating a ratio based on a comparison between a crude rate derived from person-years since leaving service in one population, and an annual incidence in another population, must be in question. 

Similar concerns apply to Figure 1, which also presents rate per 100,000 person-years at risk, by years since discharge. The number of person-years at risk (and hence the denominator) inevitably rises with years since discharge, so the methodology may have resulted in this important figure concealing the true picture. Again, there is an inconsistency; the median interval between discharge and death was 9 years (from the second paragraph of the Results) but that is in no way reflected in Figure 1.

Overall, it is suggested that survival analysis (Cox proportional hazard analysis) would be a better methodology for assessing risk of suicide, using age as the time-dependent covariate instead of time since leaving service, and using age at suicide as the failure time. A comparison group of non-veterans could be selected, which would allow valid comparison with the general population. Calculation of hazard ratios for a wide range of subgroups of interest would be facilitated, particularly for those putative risk factors where a general population comparator was appropriate. 

In specific comments:

Lines 19 and 20 of the Abstract state that there have been no recent UK studies investigating veteran suicide. It would be more accurate to say "no recent UK-wide studies" since Reference 11 is a study of Scottish veterans published in 2022.

Line 88 refers to the 2011 Armed Forces Covenant. For completeness, and to bring the Government perspective on caring for veterans up to date, the 2018 Veterans' Strategy and 2022 Veterans' Strategy Action Plan should also be mentioned here.

The authors should consider noting (either in line 94 of the Introduction and/or under Limitations) the marked geographical variation in suicide rates in England and Wales as reported in Reference 34, ranging from 6.6 per 100,000 in London to 14.1 per 100,000 in the North-East, and should discuss the implications for interpretation of their results.

Line 117: Add at end of sentence "in accordance with the UK definition of a veteran".

Line 130: The comma between staffing and surveys should be removed. 

Line 189: The description of the Cox proportional hazard analysis should specify the time-dependent covariate and failure time.

Line 240: The words "If anything" should be replaced with "Overall".

Line 392: It would be worth clarifying " . . veterans who had served solely as reserves . ." as a number of people go on from Regular service to join the Reserves and presumably they would not have been excluded from consideration as Regulars.

Under Limitations, the lack of adjustment for socio-economic status should be mentioned, as should the lack of adjustment for geographical variations in background population suicide rate, as noted above. The relatively short length of the study, limiting the number of veteran suicides able to be included in the analysis, should also be mentioned here, albeit it was constrained by the availability of relevant data as noted in the Methods.

*** Reviewer #2: 

[See attachment]

Michael Dewey

*** Reviewer #3: 

Thank you for inviting me to review this interesting paper and congratulations to the authors for producing a very interesting read, as well as additional pieces to the puzzle of suicide in servicemen and women.

MAJOR COMMENTS

1. ALL CAUSE MORTALITY AND KAPLAN-MEIER CURVES

It would be very interesting to see data for all-cause mortality as well as Kaplan-Meier curves for suicide (within the UKAF population) stratified by the predictors.

2. DEPLOYMENT

Isn't it highly likely that this is a double healthy worker effect, that the servicemen who are selected to be deployed go through additional physical and psychological testing before being deployed, and therefore cannot be expected to be comparable to either the general population or the non-deployed population (where rejects also are included)? This has been found in Scandinavian deployed veterans, who have lower suicide risk than age-sex-matched general population but similar suicide risk when matched to comparators of similar intelligence, psychological evaluation scores and other physical and mental attributes (Pethrus et al [Sweden], and Pethrus et al [Sweden, Norway, Denmark], 2022).

3. YOUNG AGE AS RISK FACTOR

It would be interesting to see more discussion about this group in terms of selection. Many of these young servicemen appear to have been discharged for medical reasons, discipline or admin reasons. May these early discharges be indicators of not well-functioning young men? While the protective effects seen among the servicemen >35y may be an indicator of well-functioning individuals, as evidenced by their ability to function for a longer period in their job?

4. MENTAL HEALTH COHORT: Comparison numbers for health care contacts by age?

It is described that UKAF and GenPop had similar contact% with mental health services (25% and 27%). For the youngest age groups, I find only data for one group (I guess the UKAF group; 10% for 16-19y and 23% for 20-24y). It would be nice to see both UKAF and GenPop data for these age groups.

5. CAUSAL LANGUAGE

Causal language is often used such as "increased the risk of suicide". It may be better to use non-causal language such as "was associated with higher/lower risk".

*** Reviewer #4: 

Overall, this is a welcome, well constructed paper which covers an important topic and uses high quality data. It is definitely worthy of publication with some minor amendments as below.

In then abstract it is unclear to me why it is not made clear that the SMR was lower for all veterans than the general population (as CIs do not include 100). This should be corrected. In the financial disclosure section any of the authors are paid (salary) by the MOD should be disclosed (e.g. KH) . The same should go for authors who are paid by other government departments (e.g. AB).

In the introduction, it would be helpful to mention the Contact Group and the RC Psych Quality Improvement for Veterans MH services as initiatives which also have aimed to improve the care available for veterans [given that this topic is discussed].

In the study setting it would be helpful to explain that the one day post service was used as this is the accepted UK definition - otherwise readers who do not know this might be perplexed why this definition was used.

why was data on deployment to Northern Ireland not included. Many of those who might have died by suicide could well have served in what was a very challenging operation. This should be corrected if at all possible. Also important, but perhaps less so, are the operations in the former Yugoslavia. Again, this was a major campaign and should be included.

With the controls, might they have served as a reserve? If so, can reserves be excluded as comparators.

In the results - Officers and NCOs should not be conflated as Officers. If needed saying NCO or Commissioned Officers or rank higher than Corporal or equivalent would be better

In the discussion - again there have been improvements in the NHS care delivered but also matching improvements with charitable care which now has a QI (RC Psych) kitemark and a wider range of charities delivering NICE care (Contact has helped with this).

The 4 years to seek help is at odds with a paper looking at Combat Stress attendees which found that it was 2 years to seek help (van Hoorn LA et al, 2013)

***

[LINK]

---

## [Decision Letter · Decision Letter 2]

31 May 2023

Dear Dr. Rodway,

Thank you very much for re-submitting your manuscript "Suicide after leaving the UK Armed Forces 1996-2018: a cohort study" (PMEDICINE-D-22-04050R2) for consideration at PLOS Medicine.

I have discussed the paper with editorial colleagues and it was also seen again by four reviewers. I am pleased to tell you that, provided the remaining editorial and production issues are fully dealt with, we expect to be able to accept the paper for publication in the journal.

[LINK]

Please let me know if you have any questions, and we look forward to receiving the revised manuscript.   

Sincerely,

Richard Turner PhD

Consulting Editor, PLOS Medicine

plosmedicine@plos.org

Requests from Editors:

At line 20 (abstract), you might wish to add a few words to explain the interest in exploring the research question now, given recent military engagements. 

At line 24, please add a sentence, say, to summarize the statistical methodology used. 

Around line 24, we suggest quoting some additional information, such as the number of person-years at risk, the breakdown of the cohort by sex and age and the median length of follow-up. 

At line 41, please soften the wording by adapting the text to "In this study, we found that suicide risk in personnel leaving the UK armed forces was not high ..." or similar. 

At line 42 and any other instances, please adapt the text to "... were associated with elevated risk ..." or similar. 

At line 43 and any other instances we suggest minor rewording so that "We/Our" is reserved for the authors, the text here becoming, e.g., "The focus ...".

We suggest using the active voice in a few elements of the Author summary, and suggest "We linked ..." at line 57.

At line 151, please correct "STOBE".

At line 300 you quote a raised suicide risk "two to three times higher" in younger groups. Is there an issue of consistency with findings quoted in the abstract here? Please correct if so. 

Noting the word "gender" at line 426, and any other instances, would "sex" be more appropriate?

At line 575, we suggest "... and this is roughly equivalent ...".

Comments from Reviewers:

*** Reviewer #1: 

[See attached]

*** Reviewer #2: 

The authors have addressed all my points. I had not realised quite how much of a disclosure risk would be involved in presenting the data for women. The authors are of course correct.

Michael Dewey

*** Reviewer #3:

Thank you for a thorough revision and well-motivated answers to the reviewer comments. I only have minor comments at this stage and look forward to hopefully seeing this interesting paper out in the public domain soon.

MINOR COMMENTS

ABSTRACT CONCLUSION

"Conclusions Suicide risk in veterans is not high but there are important differences according to age, with higher risk in young men and women. We found a number of factors which increased the risk of suicide but deployment was associated with reduced risk."

COMMENT: Causal language is still used ("increased the risk" and "reduced risk"). I would suggest using HIGHER instead of "increased" and LOWER instead of "reduced".

DISCUSSION

MAIN FINDINGS: "Methods of suicide in veterans were comparable to the general population, although hanging and strangulation were more common (62% versus 44%) and self-poisoning less common (14% versus 24%) in veterans - potentially reflecting the gender composition of the veteran cohort compared with the general population."

Are there not GenPop statistics split by sex so that a comparison of male veterans with men in the general population can be done?

STRENGTHS AND LIMITATIONS: "... may continue to improve health outcomes and reduce suicide risk in this vulnerable population."

QUESTION: What vulnerable population is referred to - is it the young ones who were discharged early? Because overall the veterans do not seem to be a vulnerable population, judging from the lower SMR.

*** Reviewer #4: 

accept

***

[LINK]

---

## [Editor Report · Decision Letter 3]

6 Jul 2023

Dear Dr Rodway, 

On behalf of my colleagues and the Academic Editor, Dr Bergman, I am pleased to inform you that we have agreed to publish your manuscript "Suicide after leaving the UK Armed Forces 1996-2018: a cohort study" (PMEDICINE-D-22-04050R3) in PLOS Medicine.

Prior to final acceptance, please also address the following issues:

At line 25 (abstract), please add "We carried out a retrospective ..." or similar at the start of the sentence;

At line 46 (abstract) please amend the tense for consistency ("... was not high but there were important ...");

We suspect that "Sex" should be substituted for "Gender" in table 2;

Our academic editor commented: "In the S1 supplementary figure, graph S1c appears to show the Deployed subgroup having a poorer outcome than the Non-deployed subgroup (compare S1b Trained vs. Untrained), which is not consistent with the hazard ratio presented at Table 2. Is the figure legend correct or have the colours/labels been inadvertently swapped? It is unfortunately too easy to do that when preparing a Stata graph for publication".

PRESS

Sincerely, 

Richard Turner PhD

Consulting Editor, PLOS Medicine

plosmedicine@plos.org